**Data Availability Statement:** The data used in this study is available. We are suggesting Smart

# HIV, syphilis and sexual-risk behaviours' prevalence among in-and out-of-school adolescent girls and young women in Zambia: A cross-sectional survey study

Patrick Musonda[1]*, Hikabasa Halwiindi[1], Patrick Kaonga[1], Alice Ngoma-Hazemba[1], Matildah Simpungwe[1], Mable Mweemba[1], Chowa Tembo[2], Cosmas Zyambo[1], John Chisoso[1], Margaret Munakampe[1], Powell Choonga[2], Owen Ngalamika[3], Mwiche Musukuma[1], Malizgani P. Chavula[1], Noah Sichula[4], Oliver Mweemba[1], Joseph Mumba Zulu[1], Henry Phiri[2]

1 School of Public Health, University of Zambia, Lusaka, Zambia, 2 Ministry of Health in Zambia, Lusaka, Zambia, 3 School of Medicine, University of Zambia, Lusaka, Zambia, 4 School of Education, University of Zambia, Lusaka, Zambia

* pmuzho@hotmail.com

## Abstract

### Background

In Zambia, 3.8% of young women and men aged 15–24 are HIV positive. However, like in most developing nations, HIV prevalence is higher among young women than young men (5.6% versus 1.8%). Despite the recognition of the rights of young people to sexual reproductive health (SRH) information and services, adolescent and young people (AYP) still face challenges in accessing healthcare in public health institutions including access to comprehensive knowledge on HIV/AIDs, HIV testing and contraceptives. The overall objective of the study was to collect baseline HIV, SRH and gender based violence (GBV) data at district level to inform the design of interventions targeting adolescent girls and young women (AGYW) aged 10–24 years in 20 districts of Zambia.

### Methods

A cross-sectional, mixed-methods study was conducted in 20 districts of Zambia with the highest incidence of HIV. Data was collected between August and October 2022 with a total response rate of 92% (12,813/13960), constituting 5979 (46.7%) in-school and 6834 (53.3%) out-of-school participants.

### Results

Overall, Mwinilunga, Chinsali, Chisamba and Chembe districts had the highest number of respondents, while Sinazongwe and Mungwi districts contributed the least. The overall age distribution was such that 12.6% (n = 1617) of those interviewed were aged 10 to 14 years, 35.4% (n = 4536) were aged 15–19 years, and 52.0% (n = 6660) were aged 20–24 years. The overall mean age at first sex among AGYW interviewed was 16.6 years which was

Zambia, a government agency that will have access to the MoH server for data and the contact will be the Director ICT Mr Andrew Kashoka; email andrewkashoka@gmail.com.

**Funding:** This study was funded by the Global Fund. The Zambia AGYW program is part of the Global Fund Strategy (2017-2022) to reduce new HIV infections among AGYW by 58% by 2022 in13 sub-Saharan African countries including Botswana, Cameroon, Kenya, Lesotho, Malawi, Mozambique, Namibia, South Africa, Eswatini, Tanzania, Uganda, Zambia and Zimbabwe. However, the funders played no part in writing of this manuscript, any errors or mistakes are thoroughly the responsibility of the authors.

**Competing interests:** No authors have competing interests.

broken down as follows: 16.2 years for in-school and 16.8 years for out of school. Overall, most of the respondents had first time sex with either their boyfriend (80.4%) or husband (15.6%), with 2.4% of the in-school participants reporting to have had their sexual debut in marriage compared to 21.0% among out-of-school AGYW. Prevalence of HIV was higher in the out-of-school compared to the in-school participants (5.5% vs 2.0%), Similarly, the prevalence of syphilis was higher in the out-of-school than the in-school participants (4.1% vs 1.5%).

## Conclusion

The study focused on assessing the prevalence and vulnerability of HIV, syphilis, GBV, and SRH services uptake among adolescent girls and young women, and exploring factors affecting girls' stay-in-school and re-engagement. The study found that HIV and syphilis are still significant public health problems among adolescent girls and young women in Zambia, emphasizing the need for increased efforts to prevent and manage these infections.

## Introduction

Adolescence is a transition period in life when an individual is no longer a child but not yet an adult. The WHO defines adolescents as individuals in the 10-19-year age group and "youth" as the 15-24- year age group. These two overlapping age groups are combined in the group "young people ", covering the age range 10–24 years. Choices made during adolescence not only have immediate consequences but also greatly influence the economic opportunities, health outcomes, and skill sets obtained later in life [1]. In 2019, 16% (1.2 billion) of the world's population was estimated to be aged between 15–24 years. Two hundred eleven million (211,000,000) of these are in sub-Saharan Africa and make up 23% of the region's population. In Zambia, adolescent girls, and young women (AGYW) account for 17% of the country's population which is estimated to be over 18 million in 2021 [2]. Adolescent and young girls' needs are not homogenous and differ with stage of development, life circumstances and the socio-economic conditions of their environment. The unique contexts in which many young people in low-income and middle-income countries are developing necessitate a deeper understanding of the issues affecting their health and wellbeing Blum et al., [3].

AIDS remains the fourth leading cause of death among individuals aged 10–19-years in African low- and middle-income countries (LMICs), despite improvements in detection, treatment, and care. Young people are vulnerable to HIV at two stages of their lives; early in the first decade of life when HIV can be transmitted from mother-to-child, sometimes known as vertical transmission, and the second decade of life when adolescence brings new vulnerability to HIV. Every week, around 5000 young women aged 15–24 years become infected with HIV (AIDS statistics| UNAIDS (2020)) [4].

In Zambia, 3.8% of young women and young men aged 15–24 are HIV positive (Ministry of Health, 2020) [5]. However, like in most developing nations, HIV prevalence is higher among young women than young men (5.7% versus 1.8%) (Ministry of Health, 2020 [5]). The 2022 Zambia Population-based HIV Impact Assessment (ZAMPHIA) [6] estimated the annual HIV incidence among young person's 15–24 years to be 0.57% and this is disproportional between males and females i.e., 0.08 among males and 1.07 among females. Similarly, HIV prevalence was estimated to be disproportional at 2% among males 20–24 years compared to

8.3% among females in the same age group. This suggests a need for more intensive primary prevention among HIV-negative women as well as targeting of secondary prevention, including safer sexual behaviours, HIV diagnosis, and treatment among HIV-positive individuals [5].

Nakazwe et al., (2022) [7] found that, HIV infection among young people in Zambia is more strongly associated with individual-level socioeconomic factors compared to neighbourhood factors. Individua-level education remains an important socioeconomic factor associated with reduced odds of HIV infection. This suggests that the HIV response in Zambia should still focus on individual level prevention strategies.

Other studies have also documented the association between schooling status and HIV infection. A study using Demographic and Health Survey data from nine DREAMS (Determined, Resilient, Empowered, AIDS-free, Mentored, and Safe) countries in Eastern and Southern Africa (Lesotho, Eswatini, Uganda, Kenya, Malawi, Mozambique, Tanzania, Zambia and Zimbabwe) found that being currently in school was associated with reduced odds of HIV infection among women aged 15–19 years in four of the nine countries (Lesotho, Eswatini, Uganda and Zambia); however, there was no significant association between being in school and HIV infection in five of the nine countries (Kenya, Malawi, Mozambique, Tanzania and Zimbabwe) [8].

In Zambia, the Technical Education, Vocational and Entrepreneurship Training (TEVET) system comprises the formal, informal and non-formal learning which enrols learners from the school system, including dropouts and the never been to school (TEVET, 2020) [9]. Vocational training institutions offer an opportunity to girls who may not have finished secondary education to obtain skills and trades and make them employable. Young women subjected to vocational training usually initiate their own projects and score significantly higher on a business knowledge index than young women not subjected to vocational training. Ensuring that classes and vocational training are accessible by women is crucial to the success of women's economic empowerment. Literature shows that improving women's socioeconomic status in society reduces their risk of contracting HIV because they are less likely to engage in risky sexual behaviour like transactional sex Haberland et al., 2021 [10].

Matovu et al. (2021) [11] reported a study conducted among young women aged 13–23 years in rural South Africa which found that over a period of 3.5 years of follow-up, the cumulative incidence of HIV was 19.9% among young women with low school attendance (<80% school days) [12]. The same study reported a study by Stoner et al, 2017 [12] who found that the cumulative incidence for herpes simplex virus type 2 (HSV-2) followed a similar trend in that 38.5% of young women with low school attendance had HSV-2 at the end of the follow-up period versus 17.3% among those with high school attendance. Further, the study [12] showed the weighted hazard of HIV and HSV-2 was greater for young women who attended less school than those who attended more school and among those who dropped out than those who stayed in school.

A good number of studies have corroborated the findings conducted in South Africa and Zimbabwe which showed that out-of-school young women had three or more times higher odds of HIV or HSV-2 infection than those who were in school [13–15]. Hence, it is undoubted that keeping girls in school is crucial for improving their health outcomes, although further research is still needed to improve our understanding of differences in risk-taking behaviours and the prevalence of HIV and other sexually transmitted infections (STI) between in-and out-of-school Adolescent Girls and Young Women (AGYW).

Although there is preponderance of evidence to confirm the association between schooling status and the risk of HIV infection [12–14, 16, 17], several studies did not include both behavioural and biomarkers in the same study while schooling status was defined using a self-

reported questionnaire on highest level of education attained. On the other hand, some studies enrolled older adolescent girls aged (15–19) or young women (15–24) but did not include the very young adolescents 10–14. Similar to a study conducted in Uganda by Matovu et al. (2021) [11], we assessed sexual-risk behaviours and HIV and Syphilis prevalence among currently in-school and out-of-school AGYW aged 10–24 years. Similarly, we were also interested to bridge the gap that exist in a number of studies in targeting the very young adolescents who are vulnerable to misinformation on sexual health matters and are at increased risk of HIV and other sexually transmitted infections (STIs).

## Materials and methods

### Study site

This study was part of a large baseline survey to collect information on HIV, Sexual Reproductive Health (SRH) and Gender Based Violence (GBV) at district level. The study was designed to inform the design of interventions targeting AGYW aged 10–24 years in 20 target districts from all ten provinces of Zambia. Data was collected from: Central Province (Kabwe, Chisamba, and Mkushi), Copperbelt Province (Chililabombwe, Ndola, Kalulushi), Eastern Province (Petauke, Nyimba), Luapula Province (Mwense, Chembe), Lusaka Province (Chongwe), Muchinga Province (Chinsali), Northern Province (Kasama, Mungwi), North-Western Province (Mwinilunga), Southern Province (Sinazongwe, Kalomo, Mazabuka), Western Province (Limulunga, Mongu), Fig 1 below shows the provinces and districts selected in the study.

The Zambia AGYW program is part of the Global Fund Strategy (2017–2022) to reduce new HIV infections among AGYW by 58% by 2022 in13 sub-Saharan African countries

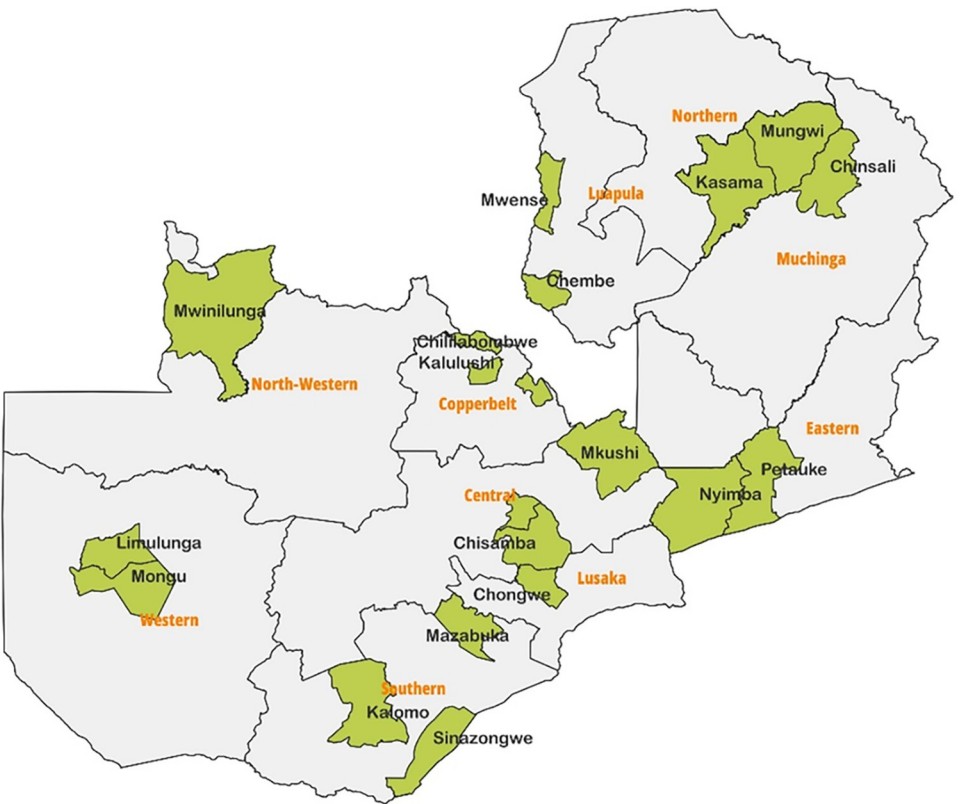

**Fig 1. Provinces and districts of the study area in Zambia.**

including Botswana, Cameroon, Kenya, Lesotho, Malawi, Mozambique, Namibia, South Africa, Eswatini, Tanzania, Uganda, Zambia and Zimbabwe [18, 19].

## Data collection procedures and methods

Data was collected using **three data collection methods**:

a. extensive desk/documents/literature reviews;

b. secondary data analyses and

c. primary data collection as described below.

*a) Extensive desk/documents/literature review*: Prior to data collection, we conducted a review of existing published (and unpublished) data on adolescent girls and young women's access to and utilization of HIV and sexual and reproductive health services. The key documents reviewed included:

- Reports from key implementing partners targeting adolescent girls and young women (e.g. DREAMS, Churches Health Association of Zambia, Discover health, etc.)

- Zambia Demographic & Health Survey Report (2007, 2013/14, 2018)

- Zambia Population-based HIV Impact Assessment Report (2016)

- Joint UN Program on HIV/AIDS reports

- Annual reports from UNFPA and UNICEF

- Annual performance reports from the Ministry of Health (with a specific focus on sexual and reproductive health)

- National Strategy on Ending Child Marriage in Zambia 2016–2021

- Zambia National Health Strategic Plan 2017–2021 [5]

- Reports from the Ministry of Education and Ministry of Community Development and Social Services among others.

- Adolescent Health Strategy 2017–2021

We also reviewed published papers on adolescent girls and young women regarding their access to and utilization of HIV, GBV and sexual and reproductive health services [10].
The desk review that we undertook helped to:

- Clarify the context/scope of the study and the gaps in the existing information in relation to the proposed objectives.

- Provide a guideline on what primary information is required and needs to be collected during the subsequent phases.

- Guide in developing instruments to collect adequate data for this exercise.

- Provide a situational analysis of what was happening currently as documented on girls and women in- and out-of-school, second chance interventions/programs to get back girls and women out of school back to school, facilitators and barriers to access and uptake of HIV/ SRH services as they related to population growth and market access approaches.

The desk review we proposed in this study was implemented as follows:

Purposive sampling was used to identify the relevant documents that were to be reviewed for the study. At the national level, the study team reviewed different documents including; the Adolescent Health Strategic Plan 2011 to 2015, the National Standards and Guidelines for Youth Friendly Health Services and the Reproductive, Maternal, Newborn, Child and Adolescent Health and Nutrition Communication and Advocacy Strategy 2018–2021. Additional Information was collected from the Ministry of Health and NGOs providing SRH services to adolescents and young people in Zambia.

*b) Secondary data analysis*: Secondary analysis of Zambia Population-based HIV Impact Assessment (ZAMPHIA 2016) and Zambia Demographic and Health Survey (2007, 2013/14, 2018) [20] was conducted to generate trends in HIV prevalence and sexual risk behaviours (e.g. transactional sex, teenage pregnancy and motherhood, having multiple sexual partners, and non-use of condoms in high-risk sexual relationships); health-seeking behaviours, access to HIV services (e.g. HIV counselling and testing, pre or post -exposure prophylaxis, condom promotion and provision, and sexually transmitted infections (STI) screening and treatment) and access to SRH services including post-abortion care, contraceptive services among AGYW.

*c) Primary data collection*: This comprised of administration of survey questionnaires (quantitative interviews) to randomly selected adolescent girls and young women living within the selected districts. All AGYWs enrolled for the quantitative component of this study were subjected to HIV and syphilis testing to determine the sero-prevalence of the two diseases. Besides the quantitative interviews, focus group discussions, in-depth interviews and key informant interviews were conducted with purposely selected respondents to explore facilitators, opportunities and barriers to utilization of HIV, GBV and SRH services among AGYW, among other aspects. The data collection methods varied by each objective. However, in this paper, we only concentrate on the quantitative findings.

## Study design and population

This was a cross-sectional, mixed-methods study. A convergent research design was employed to combine the qualitative and quantitative data collected among the in-and-out-of-school AGYW aged 10–24 years across the 20 districts. This implies that data collection and analysis of both quantitative and qualitative data was conducted simultaneously and results were integrated in the interpretation phase (Moseholm and Fetters, 2017) [21]. In-school AGYW were those that were currently in school at the time of the survey while out-of-school AGYW were those dropped out of school prior to school completion or adolescents and young women who were out of formal education or enrolled in a vocational training institution at the time of the interview. Hence, this excluded enrolling out-of-school AGYW who were not in school at the time of the survey because they completed school.

The study population included in-school and out-of-school adolescent girls and young women aged 10–24 years, resident in the selected districts. Besides AGYW, we also interviewed selected key informants involved in HIV, GBV and SRH programming including implementing partners. A list of key informants is listed in Table 1 below.

**Table 1. List of major sources of key informants.**

| | |
|---|---|
| Government Departments<br>Ministry of Health (District, Health Facility, NHCs)<br>Ministry of Education<br>Ministry of Local Government<br>Community Development & Social Welfare<br>Victim Support Unit<br>One Stop Centre | • Community leaders: Traditional, PTA and Religious Leaders NGOs: Kwatu- ZCCP, CoAG, CiHeb, YWCA, DREAMS, **NZP+,** DAPP<br>• Financial Institutions: ZANACO, AGORA micro finance<br>• Trades and Vocational Training Colleges<br>• IDI, adolescent girls, sex workers (in an out of school) |

## Sample size determination

Using the prevalence formula, a sample size calculation showed that a total of 12560 was needed to detect an effect size if present in the target population. The sample size was calculated per age group using the total projected population for the year 2020 and the HIV prevalence as estimated by the Zambia Population-based HIV Impact Assessment (ZAMPHIA) [6]. The population projections were obtained from the Central Statistical Agency website. To obtain the sample size per age group, the following formula for sample surveys was used Lwanga and Lemeshow. [22].

$$n = \frac{Z^2_{1-\alpha/2}P(1-P)N}{d^2N + Z^2_{1-\alpha/2}P(1-P)}$$

Where;

P is the anticipated prevalence of the study outcome (here we used the estimated HIV prevalence among in target population (Ministry of Health) [5]), N is the target population size in each age group, d is the precision set at 0.8% (taking into account a Bonferroni correction) and Z is z-score set to 1.96 at α = 5% significance level.

To correct for non-response, the estimated sample size in each district and in each age-group was adjusted using the formula below

Adjusted Sample size = $\frac{n}{1-\left(\frac{n}{100}\right)}$

Where;

- n is the calculated sample size in each age-group and district and,

- $\frac{n}{100}$ is the anticipated non-response rate (10%).

After adjusting for non-response, total sample size was 13960.

The calculated sample size of 13,960 (Table 2) was then allocated among the in-school and the out-of-school using estimates national level estimates from the 2015 Zambia Living Conditions Monitoring Survey (LCMS) on the proportion of females in school and out-of-school. Using the 2015 LCMS, we assumed 84.8% of 10–14-year-olds, 73.4% of 15–19-year-olds and 22.5% of 20–24-year-olds were in school.

However, we managed to collect a total of 12, 813 (corresponding to 92% response rate) constituting 5979 (46.7%) in-school and 6834 (53.3%) of out-of-school participants. This was slightly above the estimated minimum sample size for the study.

## Sampling procedures

The sampling procedures was done with respect to addressing the following study objectives:

### (a) Objective 1: To determine socio-demographic, sexual, health and behavioural status among adolescent girls and young women in targeted districts in Zambia

**Approach.** This was a primary objective for collecting baseline data on the socio-demographic, sexual and health behaviour of adolescent girls and young women in the targeted districts. The sample size for this objective was, therefore, the overall sample size estimated to ensure adequate statistical power to determine district-specific parameters to inform programming. Data was collected on the socio-demographic characteristics of the respondents (e.g. age, education, occupation), sexual risk behaviours (e.g. age at sexual debut, age at first marriage (if married), number of sexual partners, condom use), health seeking behaviours (e.g. HIV counselling and testing, STI screening and treatment) and behavioural characteristics–

**Table 2. Sample size determination by selected district.**

| | 10–14 years | | | 15–19 years | | | 20–24 years | | | |
|---|---|---|---|---|---|---|---|---|---|---|
| DISTRICT | Total | In-School | Out-of-School | Total | In-School | Out-of-School | Total | In-School | Out-of-School | Total |
| Kabwe | 60 | 51 | 9 | 190 | 139 | 51 | 448 | 101 | 347 | 698 |
| Chisamba | 60 | 51 | 9 | 190 | 139 | 51 | 448 | 101 | 347 | 698 |
| Mkushi | 60 | 51 | 9 | 190 | 139 | 51 | 448 | 101 | 347 | 698 |
| Chililabombwe | 60 | 51 | 9 | 190 | 139 | 51 | 448 | 101 | 347 | 698 |
| Ndola | 60 | 51 | 9 | 190 | 139 | 51 | 448 | 101 | 347 | 698 |
| Kalulushi | 60 | 51 | 9 | 190 | 139 | 51 | 448 | 101 | 347 | 698 |
| Petauke | 60 | 51 | 9 | 190 | 139 | 51 | 448 | 101 | 347 | 698 |
| Nyimba | 60 | 51 | 9 | 190 | 139 | 51 | 448 | 101 | 347 | 698 |
| Mwense | 60 | 51 | 9 | 190 | 139 | 51 | 448 | 101 | 347 | 698 |
| Chembe | 60 | 51 | 9 | 190 | 139 | 51 | 448 | 101 | 347 | 698 |
| Chongwe | 60 | 51 | 9 | 190 | 139 | 51 | 448 | 101 | 347 | 698 |
| Chinsali | 60 | 51 | 9 | 190 | 139 | 51 | 448 | 101 | 347 | 698 |
| Mungwi | 60 | 51 | 9 | 190 | 139 | 51 | 448 | 101 | 347 | 698 |
| Kasama | 60 | 51 | 9 | 190 | 139 | 51 | 448 | 101 | 347 | 698 |
| Mwinilunga | 60 | 51 | 9 | 190 | 139 | 51 | 448 | 101 | 347 | 698 |
| Sinazongwe | 60 | 51 | 9 | 190 | 139 | 51 | 448 | 101 | 347 | 698 |
| Mazabuka | 60 | 51 | 9 | 190 | 139 | 51 | 448 | 101 | 347 | 698 |
| Kalomo | 60 | 51 | 9 | 190 | 139 | 51 | 448 | 101 | 347 | 698 |
| Limulunga | 60 | 51 | 9 | 190 | 139 | 51 | 448 | 101 | 347 | 698 |
| Mongu | 60 | 51 | 9 | 190 | 139 | 51 | 448 | 101 | 347 | 698 |
| TOTAL | | | | | | | | | | 13960 |

including knowledge and use of contraception, knowledge of the menstruation cycle and how AGYW manage their menstrual periods, and source of information on HIV and SRH issues, among others.

**Overall sampling procedures.** In the first stage, a total of 540 enumeration areas (EAs) were selected, that is, 28 EAs from each selected district. To randomly select the EAs, a sampling frame based on the 2010 Census of Population and Housing of the Republic of Zambia, provided by the Zambia Statistics Agency was used. An enumeration area (EA) is a geographical area consisting of an adequate number of households and serves as a counting unit for the population census. Each EA has a sketch map delineating its boundaries, with identification information and a measure of size, which is the number of residential households enumerated in the 2010 Census (Zambia Statistics Agency *et al.*, 2019) [23]. A household listing exercise was then conducted in each selected EA. The objective of the listing exercise was to create a listing frame from which the participants were to be randomly selected. The listing exercise comprised enumerators collecting data on age and schooling status on each adolescent and young person in each household in the selected EA. Participants were then randomly selected from the sampling frame created during the listing exercise.

**(b) Objective 2: To determine the prevalence of HIV and Syphilis among adolescent girls and young women in the targeted districts**

All (12,813) AGYW who participated in the structured questionnaires interviews were eligible and participated in the HIV and Syphilis testing. All adolescents who consented to the HIV and Syphilis testing were provided with pre-test and post-test counselling by trained counsellors and were tested by trained laboratory scientists to determine the prevalence of these STIs

in this population. The participants could opt out of taking the HIV or Syphilis test or both. Opting out meant also dropping out of the study.

**Specific objectives were:**
To carry out a baseline survey:

a. To determine the sero-prevalence of HIV and syphilis

b. To determine the prevalence of reported symptoms and signs of STIs

Upon completion of the survey and administration of additional informed consent procedures, adolescents were requested to voluntarily provide a blood sample for testing for HIV and/or syphilis.

Blood samples were collected at the location of the interview, that is, at the participants' place of residence. To collect and test blood samples, two laboratory scientists and two HIV counsellors affiliated with a health facility within the district were included on each field team. To obtain informed consent for blood collection, the laboratory scientists explained the procedure of the blood test and the confidentiality of the test results. The laboratory scientists were also informed of the participants' right to opt out of the HIV or Syphilis test or both tests. The participants were also informed of their right to have their results given to them or not. Counselling before and after the blood tests were provided by the HIV counsellor.

For samples requiring further analysis, laboratory scientists asked for permission to store left over blood for future unspecified tests. For non-emancipated respondents, age 10–17 (i.e., those who still lived with their parents/guardians), laboratory scientists also sought consent from their parents or guardians. After obtaining consent and assent, laboratory scientists proceeded to draw blood from the finger and/or arm by venipuncture using an evacuated tube collection system.

**Linkage to care among AGYWs confirmed to be HIV or syphilis positive, and AGYW reporting gender-based violence.** All AGYWs confirmed HIV-positive were referred to the nearest health facility for enrolment into HIV care as per existing enrolment guidelines. In addition, those who tested HIV negative but were syphilis positive were referred to the nearest health facility for further management. Individuals that report gender-based violence were referred to the existing gender-based violence programs in the community for better management.

**(c) Objective 3: To conduct a needs assessment to establish the vulnerability, wealth, and comprehensive HIV knowledge index of adolescent girls and young women in the targeted districts**

**Approach.** To create the Adolescent Girls Multilevel Vulnerability Index (AGI), we followed a four-step process as outlined in the Handbook on Constructing Composite Indicators (OECD 2008) [24]. The first step was the development of a theoretical framework to inform selection of indicators into a composite score. We matched the framework with the available data sources and indicators to guide us in constructing a composite index. Lastly, we tested the validity and robustness of the vulnerability index through a sensitivity analysis. We illustrate each step in detail below.

*Step 1: Developing the theoretical framework.* For the purpose of the index, we followed the United Nations definition of adolescence, which refers to individuals between the ages of 10 and 19 years (UNICEF 2003) [25]. The AGI is focused on adolescent girls rather than boys because of the unique risks faced by girls in this age group. Consequently, we concentrate on the girl child in this study. The indicators included in the index were based on research and

expertise on adolescent girls. Risks such as child marriage, limited economic opportunities, and parenthood have different effects on the outcome of the adolescent girls. Vulnerability was defined as a relationship between the context in which a girl lives and a set of factors that put her "at-risk" for negative outcomes. These factors are better explained by socio-ecological model and are presented at multiple levels and (Fischhoff, Nightingale, and Iannotta 2001) [26], including the level of the:

a.  individual girl (e.g., a chronic health condition like asthma), genetic (e.g., predispositions to disease), and psychological (e.g., thoughts, emotions, and behaviours);

b.  her proximal social settings, (such as familial structures (e.g., orphan), peer networks (e.g., older, more experienced peers), classrooms (e.g., teachers who do not show up for work), and community characteristics (e.g., violence, gangs)

c.  more distal levels (girl can also be influenced negatively by societal practices and beliefs e.g., female genital cutting and a policy environment that does not recognize her rights)

In addition, there was a need to promote positive interactions between the development of a young person and her context to increase the likelihood that she will thrive (Lerner et al. 2011) [27]. This calls for programs and policies to focus not only on prevention and problem reduction, but also on creating contexts to promote thriving. It has been noted that vulnerability affects girls at different levels listed above. The AGI index conceptualizes vulnerability using the multi-dimensional measures lens as well as at multiple levels of risk. Therefore, the AGI alludes to the notion that comprehensive actions are more effective than population-based actions that focus on a single issue.

*Step 2*: *Selecting the data and potential indicators*. For individual and household level indicators, we collected primary data using the quantitative questionnaire. The questionnaire collected data on the following indicators:

a.  **Individual level:** We selected the indicators relating to schooling and family structure, marital status, maternal health, high-risk sex and family formation.

b.  **Household level**: Data was collected on housing conditions and household and family structure.

For the community risk factors, we conducted secondary data analysis of the 2018 Zambia Demographic Health Survey.

a.  **Community risk factors:** Data included child marriage, illiteracy, and low comprehensive knowledge of HIV/AIDS.

*Step 3*: *Constructing the index*. **Vulnerability index.** Vulnerability was assessed as risk factors within three dimensions (individual, household and community, see Table 3 below). Data was summed to create the composite indicator with multiple cut-off points at different levels. The indicators were dichotomised and coded as 0 for "no" and 1 for "yes." Variables were standardized so that each contributed equally to the overall summary statistic. The denominator included all girls in the sample. At the individual level for girls aged 10 to 14, a girl was classified as vulnerable if they were at least two years behind grade for age or were not in school and/or not living with their parents. The cut off for individual girls ages 15 to 19 years, one was considered to have individual level vulnerability if she has ever been married, or given birth or currently married, or did not attend secondary school, or engaged in high-risk sex, that is, sex under the age of 15 or multiple non regular partners. At household level, a girl (10–19 years) was classified as vulnerable if she experienced any two of the following five conditions: no

Table 3. Categorization of vulnerability index.

| Indicator level | Age group | |
|---|---|---|
| | 10–14 years | 15–19 years |
| **Individual** | Two years behind grade for age OR no education | Currently married |
| Among 10–14: 1 of 2 | Not living with parents | Ever given birth or currently pregnant |
| Among 15–19: 1 of 4 | | Did not attend secondary school |
| | | High-risk sex: Sex under the age of 15, multiple partners or non-regular partners |
| **Household** (2 of 3) | 10–19 years | |
| | No access to an improved source of water | |
| | No access to improved sanitation | |
| | Household head has no education | |
| **Community** (1 of 3) | High rate of early marriage before 18 for women 20–49 (above mean) | |
| | High rate illiteracy in women aged 20–49 before the age of 18 (above mean) | |
| | High prevalence of HIV (% of population ages 15 to 49) (above mean) and high rate of no comprehensive knowledge of HIV for women 20–49 (above mean) | |

access to improved source of water, no access to improved sanitation, household head has no education, food insecurity (no access to food in a day), and non-family support, that is, ever consulted other for social support other than a family member. At the community level, girls were considered vulnerable if they lived in a community characterised by any one of the following: high rate of early marriage before the age of 18, high rate of illiteracy, increased prevalence of HIV, and low comprehensive knowledge of HIV. At each level, a score of 1 was given if a girl experienced these measures and 0 otherwise. We then used principal component analysis on the scored data (0/1) to derive the vulnerability index. The vulnerability index was divided into tertiles (low, medium and high) with the highest tertiles representing the most vulnerable group.

**Wealth tertile index.** This index constituted of responses on household possessions representing a wealth proxy for the AGYW interviewed. The list of household assets probed mimicked what the study by Matovu et al., 2021 [11] used by including: whether or not the respondent owned a home or lived in a family home; ownership of a radio; television set; bicycle, motorcycle, cell phone, regular (landline) phone, computer, income-generating business, indoor bathroom, running water either inside the house or inside the compound of the house, electricity, car, generator and solar electricity.

To construct the socio-economic status (SES)/wealth index, each household item was assigned a weight ascertained through principal component analysis. After that, the scores were standardised in relation to a standard normal distribution with a mean zero and standard deviation of one. For each individual, the scores on household possessions were then summed up, ranked and subdivided into wealth tertiles (low, middle, and high), depending on the scores, with each tertile containing a third of the participants.

**Comprehensive knowledge of HIV.** Similarly, the definition of Comprehensive HIV knowledge was based on the derivation by Matovu et al., 2021 [11] by using the following variables: 1) knowing that consistent use of condoms during sexual intercourse and having just one uninfected, faithful partner can reduce the risk of getting HIV; 2) knowing that a health-looking person can have HIV, and 3) rejecting the two most common misconceptions about

HIV transmission or prevention, namely: a) belief that one can acquire HIV from mosquito bites and b) belief that one can acquire HIV by sharing food with an HIV-infected person. To construct this index, responses to the above questions were assigned one and zero for a positive and negative response respectively, and a weight ascertained through principal component analysis. Then, the scores were standardised in relation to a standard normal distribution with a mean zero and standard deviation of one. For each individual, the scores on the questions were then summed up; ranked and sub-divided into three knowledge levels (low, medium, and high), depending on their scores, with each level containing a third of the participants.

## Measurement of other variables

Similar to Matovu et al. [11], the dependent variables were: 1) sexual-risk behaviour and 2) prevalence of HIV and syphilis infections, assessed separately among in-and-out-of-school AGYW. Adolescent girls and young women were deemed to have engaged in sexual-risk behaviour if they: a) reported a history of sexually transmitted infections; or b) reported that they had their first-time sexual experience before the age of 15; or c) had sexual intercourse with multiple (2+) sexual partners in the past 12 months; or d) did not use a condom or used condoms inconsistently with their most recent sexual partner. No attempt was made to create one composite variable of sexual-risk behaviours because each behaviour was considered to constitute a level of HIV/STI risk on its own. HIV and syphilis prevalence was determined as a percentage of those tested who tested positive for HIV or syphilis.

The independent variables included age-group (categorised as 10–14, 15–19, and 20–24 years), highest level of education attained at the time of the survey (in-school AGYW were asked about their current class of attendance), marital status (categorised as 'never married', 'in a relationship but not married', 'married or in union' and 'divorced/widowed/separated'), history of HIV testing (ever tested for HIV; tested for HIV in the past 12 months), alcohol use before sex, wealth tertile (categorised as 'low', 'middle', and 'high'), comprehensive knowledge of HIV (categorised as 'low', 'medium', and 'high') and vulnerability index (categorised as 'low', 'middle', and 'high'). A detailed description of how wealth tertile, comprehensive HIV knowledge and vulnerability index were derived are given in the sections above.

## Data analysis (objectives 1, 2, and 3)

Descriptive statistics such as frequencies, proportions, means, medians, standard deviation, and interquartile range were used as appropriate to summarise the characteristics of the study participants, by district, age-group, and schooling status. HIV and syphilis status were determined and presented in form of percentages, that is, frequencies and percentages of those tested for HIV and syphilis. HIV and syphilis status were cross tabulated by district, age-groups and stratified by schooling status to determine differences in the prevalence of these diseases by those characteristics. The results were presented in tables and figures as appropriate. To model possible associations between socio-demographic factors and an adolescent being HIV or Syphilis positive a logistic regression model was used.

## Ethical considerations

Ethics approval was sought from UNZABREC (REF. 2460–202) The protocol was then submitted to the Zambia National Health Research Authority (Ref No: NHRA000010/10/02/2022) for registration and Approval. All participating implementing partners (IPs) were asked for permission to have their staff participate in the study. A letter of introduction from the ministry of education and ministry of health was also obtained to ease coordination with the selected organizations and partners.

Informed consent was obtained from all the participants prior to participation, and we ensured confidentiality of data obtained. The study used written informed consent from each participant prior to data collection solicited by a trained study team member. Potential respondents were given an opportunity to ask any questions about the research before they are asked if they were willing to participate. Potential respondents who agreed to participate were interviewed. The study participants were provided with written informed consent and information participant sheets detailing the study, the risks and benefits, and emphasis on the protection of confidentiality.

For adolescents less than the age of 18 years, consent was sought from their parents for the structured interview, FGD, IDI participation and the HIV and Syphilis testing. Assent was also sought from the adolescent's 10–17 years old. Permission to conduct the structured interview, FDG or IDI and blood tests was only considered granted if both the parents/guardians and the adolescent signed the consent and assent forms respectively. However, it was also explained to the adolescents that they can opt out of the study at any time during the interview or blood test. Results from the qualitative analysis will be published in a separate paper (submitted).

A separate consent form was obtained for HIV and Syphilis testing. Adolescents who consented to participate in answering the structured questionnaire could further consent to having an HIV test and/or a Syphilis test only. Therefore, only Adolescents who consented to both the structured interview and the HIV and/or Syphilis test were tested. The results of the blood tests were not shared with anyone except the participants if requested.

To ensure privacy, HIV testing, and counselling was conducted within the dwelling or premises of the participants' residence. Adolescents who consented to the HIV and/or Syphilis test, were tested by a laboratory scientist after counselling. Counselling sessions was conducted one on one with a trained counsellor. Adolescents who participated in HIV and/or Syphilis testing were offered both pre and post HIV counselling.

## Covid prevention measures

To prevent the spread of Covid-19 among the research assistants and the participants, disposable masks were provided to both the participants and the research assistants. Before commencement of interviews, both the research assistants and the participants sanitized their hands with the alcohol-based hand sanitizer that were provided. Furthermore, a distance of at least 1 meter was maintained during the interview.

## Results

### Distribution of samples over districts stratified by in-and-out of school participants

Table 4 shows the distribution of the 12,813 (91.8% response rate) study participants. Of those interviewed, 5,979 (46.7%) were in school while 6,834 (53.3%) were out of school. The age band 15–19 had the highest number interviewed from the in-school, 3150 respondents, while the age band 20–24 had the highest interviewees among the out of school, with 5,241 respondents. Mwinilunga, Chinsali, Chisamba and Chembe districts had the highest number of respondents, while Sinazongwe and Mungwi districts contributed the least.

### Respondents' characteristics

Table 5 shows the distribution of AGYW participants by selected background characteristics. The overall age distribution was such that 12.6% (n = 1617) of those interviewed were aged 10 to 14 years, 35.4% (n = 4536) were aged 15–19 years, and 52.0% (n = 6660) were aged 20–24

**Table 4. Distribution of the 12813 individuals interviewed for the AGYW study.**

| DISTRICT | Total In-School | In-School | | | Total Out-of-School | Out-Of-School | | | OVERALL District total |
|---|---|---|---|---|---|---|---|---|---|
| | | 10–14 | 15–19 | 20–24 | | 10–14 | 15–19 | 20–24 | |
| Chembe | 289 | 52 | 140 | 97 | 412 | 9 | 50 | 353 | 701 |
| Chililabombwe | 299 | 54 | 173 | 72 | 337 | 13 | 53 | 271 | 636 |
| Chinsali | 298 | 57 | 137 | 104 | 497 | 9 | 53 | 345 | 705 |
| Chisamba | 339 | 48 | 223 | 68 | 364 | 9 | 65 | 290 | 703 |
| Chongwe | 224 | 92 | 122 | 10 | 292 | 8 | 94 | 190 | 516 |
| Kabwe | 326 | 81 | 160 | 85 | 329 | 15 | 103 | 211 | 655 |
| Kalomo | 298 | 128 | 143 | 27 | 396 | 6 | 121 | 269 | 694 |
| Kalulushi | 280 | 48 | 139 | 93 | 404 | 16 | 59 | 329 | 684 |
| Kasama | 318 | 9S9 | 136 | 83 | 348 | 11 | 66 | 271 | 666 |
| Limulunga | 370 | 103 | 187 | 80 | 269 | 8 | 61 | 200 | 639 |
| Mazabuka | 276 | 69 | 174 | 33 | 386 | 21 | 103 | 262 | 662 |
| Mkushi | 265 | 62 | 138 | 65 | 376 | 8 | 73 | 295 | 641 |
| Mongu | 352 | 56 | 234 | 62 | 288 | 3 | 72 | 213 | 640 |
| Mungwi | 248 | 51 | 165 | 32 | 220 | 7 | 55 | 158 | 468 |
| Mwense | 294 | 66 | 159 | 69 | 397 | 25 | 72 | 300 | 691 |
| Mwinilunga | 491 | 90 | 160 | 241 | 232 | 7 | 52 | 173 | 723 |
| Ndola | 310 | 81 | 153 | 76 | 342 | 8 | 59 | 275 | 652 |
| Nyimba | 231 | 56 | 140 | 35 | 403 | 12 | 55 | 336 | 634 |
| Petauke | 260 | 51 | 140 | 69 | 394 | 8 | 59 | 327 | 654 |
| Sinazongwe | 211 | 66 | 127 | 18 | 238 | 4 | 61 | 173 | 449 |
| **Totals** | **5979** | **1410** | **3150** | **1419** | **6834** | **207** | **1386** | **5241** | **12813** |

years. By school status, 46.7% (n = 5979) were in-school while 53.3% (n = 6834) were out-of-school. About half 52.1% (n = 3117) of in-school AGYW were in secondary school, 42.5% (n = 2540) were in primary school while only 5.4% (n = 322) were in vocational or tertiary institutions. For the out-of-school AGWY, 37.9% (n = 2591) attained primary education, 29.0% (n = 1982) attained secondary education while 25.5% (n = 1742) had vocational or other tertiary education.

Overall, 21.6% (n = 2765) of AGYW reported that they were married or cohabiting while 19.4% (n = 2481) were in a relationship but not married. As one would expect, a smaller proportion of AGYW of the in-school, {1.5% (n = 92)} compared to {39.1% (n = 2673)} of out-of-school AGYW reported that they were currently married or cohabiting. Slightly above three quarters 77.5% (n = 4632) of the in-school and about a third 32.8% (n = 2244) of out-of-school AGYW had never married while 1.9% (n = 114) and 8.4% (n = 577) respectively were divorced or widowed at the time of the interviews. Among those who were married or cohabiting, the age of their partners varied by schooling status: 31.1% (n = 134) of the in-school and 7.2% (n = 42) of the out-of-school were married to partners aged 15 to 19 years; 39.2% (n = 169) of the in-school and 35.6% (n = 209) out of school were married to partners aged 20 to 24 years; and, 20.0% (n = 86) of in-school and 46.7% (n = 274) of out-of-school were married to partners aged 25 years and above.

The population's ability to receive and synthesize information on matters that affect their own health was assessed by the indicator literacy levels. Literacy levels were determined by asking the AGYW to read a plain english or local language paragraph that was embedded into the questionnaire. Individuals were considered to be literate if they were able to read the paragraph with ease (with clear and correct pronunciation of words). A higher proportion 34.5%

**Table 5. Percentage distribution of the participants, overall and by school status.**

| Characteristics | Total N = 12, 813 | In-School N = 5979 (46.7%) | Out-of-School N = 6834 (53.3%) | p-value <0.0001 |
|---|---|---|---|---|
| Overall | 12, 813 (100) | 5979 (100) | 6834 (1000 | n/a |
| **Age-group (years)** | | | | |
| 10–14 | 1,617 (12.6%) | 1,410 (23.6%) | 207 (3.0%) | <0.0001 |
| 15–19 | 4,536 (35.4%) | 3,150 (52.7%) | 1,386 (20.3%) | <0.0001 |
| 20–24 | 6,660 (52.0%) | 1,419 (23.7%) | 5,241 (76.7%) | <0.0001 |
| **Education** | | | | |
| Primary | 5,131 (40.0%) | 2,540 (42.5%) | 2,591 (37.9%) | 0.0008 |
| Secondary | 5,099 (40.0%) | 3,117 (52.1%) | 1,982 (29.0%) | <0.0001 |
| More than secondary | 2,064 (16.1%) | 322 (5.4%) | 1,742 (25.5%) | <0.0001 |
| Missing/Non-response | 519 (4.1%) | 0 (0.0%) | 519 (7.6%) | n/a |
| **Religion** | | | | |
| Catholic | 2,708 (21.1%) | 1,275 (21.3%) | 1,433 (21.0%) | 0.849 |
| Anglican/Protestant | 1,228 (9.6%) | 580 (9.7%) | 648 (9.5%) | 0.905 |
| Moslem | 20 (0.2%) | 9 (0.2%) | 11 (0.2) | 1.000 |
| Pentecostal/Born Again/Evangelical | 2,946 (23.0%) | 1,329 (22.2%) | 1,617 (23.7%) | 0.336 |
| Seventh Day | 2,492 (19.5%) | 1,071 (17.9%) | 1,421 (20.8%) | 0.071 |
| Orthodox | 133 (1.0%) | 64 (1.1%) | 69 (1.0%) | 0.955 |
| Other | 3,286 (25.7%) | 1,651 (27.6%) | 1,635 (23.9%) | 0.015 |
| **Marital status** | | | | |
| Never married | 6,876 (53.7%) | 4,632 (77.5%) | 2,244 (32.8%) | <0.0001 |
| In relationship but not married | 2,481 (19.4%) | 1,141 (19.1%) | 1,340 (19.6%) | 0.754 |
| Married/cohabiting | 2,765 (21.6%) | 92 (1.5%) | 2,673 (39.1%) | <0.0001 |
| Divorced/Separated/Widowed | 691 (5.4%) | 114 (1.9%) | 577 (8.4%) | 0.015 |
| **How old is your partner** (years)* | | | | |
| 15–19 | 176 (17.3%) | 134 (31.1%) | 42 (7.2%) | 0.0019 |
| 20–24 | 378 (37.1%) | 169 (39.2%) | 209 (35.6%) | 0.472 |
| ≥25 | 360 (35.4%) | 86 (20.0%) | 274 (46.7%) | <0.0001 |
| Don't know | 104 (10.2%) | 42 (9.7%) | 62 (10.6%) | |
| **Literacy level** | | | | |
| Can read but with difficulty | 3,005 (23.5%) | 1,755 (29.4%) | 1,250 (18.3%) | <0.0001 |
| Can read with easy | 3,507 (27.4%) | 2,061 (34.5%) | 1,446 (21.2%) | <0.0001 |
| Can't read at all | 562 (4.4%) | 308 (5.2%) | 254 (3.7%) | 0.394 |
| Missing/non response | 5,739 (44.8%) | 1,855 (31.0%) | 3,884 (56.8%) | <0.0001 |
| **Who do you live with** | | | | |
| Alone | 348 (2.7%) | 73 (1.2%) | 275 (4.0%) | 0.242 |
| Friends | 107 (0.8%) | 67 (1.1%) | 40 (0.6%) | 0.793 |
| Only with mother | 2,253 (17.6%) | 1,186 (19.8%) | 1,067 (15.6%) | 0.009 |
| Only with father | 197 (1.5%) | 103 (1.7%) | 94 (1.4%) | 0.865 |
| Both parents | 4,904 (38.3%) | 3,120 (52.2%) | 1,784 (26.1%) | <0.0001 |
| With other relatives | 2,455 (19.2%) | 1,352 (22.6%) | 1,103 (16.1%) | 0.0001 |
| Husband or partner | 2,549 (19.9%) | 78 (1.3%) | 2,471 (36.2%) | <0.0001 |
| **Ever tested for HIV** | | | | |
| Yes | 7,522 (58.7%) | 2,262 (37.8%) | 5,260 (77.0%) | <0.0001 |
| No | 4,303 (33.6%) | 3,138 (52.5%) | 1,165 (17.1%) | <0.0001 |
| Missing/non response | 988 (7.7%) | 579 (9.7%) | 409 (6.0%) | 0.037 |
| **HIV test in last 12 months** | | | | |

(*Continued*)

**Table 5.** (Continued)

| Characteristics | Total N = 12, 813 | In-School N = 5979 (46.7%) | Out-of-School N = 6834 (53.3%) | p-value <0.0001 |
|---|---|---|---|---|
| No | 11,319 (88.3%) | 5,598 (93.6%) | 5,721 (83.7%) | <0.0001 |
| Yes | 1,494 (11.7%) | 381 (6.4%) | 1,113 (16.3%) | <0.0001 |
| **Comprehensive knowledge of HIV** | | | | |
| Low | 4,096 (32.0%) | 2,444 (40.9%) | 1,652 (24.2%) | <0.0001 |
| Medium | 3,464 (27.0%) | 1,738 (29.1%) | 1,726 (25.3%) | 0.012 |
| High | 5,253 (41.0%) | 1,797 (30.1%) | 3,456 (50.6%) | <0.0001 |
| **Wealth tertile** | | | | |
| Low | 43 (0.34%) | 27 (0.45%) | 16 (0.23) | 0.908 |
| Medium | 7,596 (59.3%) | 3,485 (58.3%) | 4,111 (60.2%) | 0.093 |
| High | 5,174 (40.4%) | 2,467 (41.3%) | 2,707 (39.6%) | 0.213 |
| **Vulnerability Index** | | | | |
| Low | 21 (0.16%) | 0 (0.0%) | 21 (0.31%) | n/a |
| Medium | 8,474 (66.1%) | 2,802 (46.9%) | 5,671 (83.0%) | <0.0001 |
| High | 4,318 (33.7%) | 3,177 (53.1%) | 1,141 (16.7%) | <0.0001 |
| **Condom use with most recent partner (last 12 months)*** | | | | |
| Always | 1,083 (15.0%) | 506 (25.6%) | 577 (11.0%) | <0.0001 |
| Sometimes | 2,057 (28.4%) | 661 (33.4%) | 1,396 (26.5%) | 0.0012 |
| Rarely | 793 (11.0%) | 201 (10.2%) | 592 (11.2%) | 0.695 |
| Never | 3,314 (45.7%) | 610 (30.8%) | 2,704 (51.3%) | <0.0001 |

& = z-test comparing proportions of in-school vs out-of-school

* = analysis done on complete case data excluding the missing/non response values.

(n = 2061) of the in-school AGYW were able to read the text with ease compared to 21.2% (n = 1446) of their out-of-school counterparts. Similarly, a third 29.4% (n = 1755) of the in-school AGYW were able to read but with difficulty, compared to 18.3% (n = 1250) for the out-of-school. Furthermore, only 5.2% (n = 308) of the in-school AGYW could not read at all, and 3.7% (n = 254) among the out-of-school could not read at all. These findings have implications for the use of written messages particularly among out-of-school AGYW. It should be noted that more than half of the out-of-school and 31.0% (n = 1855) in-school AGYW had missing information or non-response for this indicator.

Slightly above half 52.2% (n = 3120) of the in-school AGYW reported living with both parents compared to only 26.1% (n = 1784) of their Out-of-school counterparts. Among in-school AGYW, 22.6% (n = 1352) lived with other relatives, 19.8% (1186) with mother only and 27.9% (n = 1673) lived with either a partner, father only, other relative, friend or alone. For out-of-school AGYW 36.2% (n = 2471) lived with a partner, 16.1% (n = 1103) with other relative, 15.6% (n = 1067) with mother only, and 6.0% (n = 408) with either a friend, father only or alone.

When asked if they had ever tested for HIV, 77.0% (n = 5260) of the out-of-school said yes compared to only 37.8% (n = 2262) among the in-school AGYW. Not surprisingly, only a few of the HIV tests were done within the 12 months preceding the survey, with 6.4% (n = 381) among the in-school and 16.3% (n = 1113) among the out-of-school. Interestingly, only a third 30.1% (n = 1797) of the in-school AGYW and half 50.6% (n = 3456) of the out-of-school AGYW had high comprehensive knowledge of HIV. Similarly, 40.9% (n = 2444) of the in-school AGYW and 24.2% (n = 1652) of the out-of-school counterparts had low comprehensive knowledge of HIV. The use of condoms was low with only a quarter 25.6% (n = 506) of in-

school AGYW and 11.0% (n = 577) of out-of-school reported using condoms always. Only a third 33.4% (661) of in-school and a quarter 26.5% (1396) of the out-of-school AGYW reported using condoms sometimes with their partner.

The wealth tertile was used as a proxy of household level wealth. AGYW in the high tertile were 41.3% (n = 2467) among the in-school and 39.6% (n = 2707) among the out-of-school. Majority of the AGYW fell in the medium tertile with 58.3% (n = 34485) among the in-school and 60.2% (n = 4111) of the out-of-school AGYW. Vulnerability index was used to map out community risks, vulnerability, and susceptibility to GBV/Syphilis/HIV in the AGYW. The index is a tool that combines different indicators. High vulnerability index means a participant was very vulnerable in need of priority attention. Half 53.1% (n = 3177) of the in-school were classified as highly vulnerable compared to only 16.7% (n = 1141) among the out-of-school. Conversely, 83.0% (n = 5671) of the out-of-school AGYW were classified as having moderate vulnerability compared to 46.9% (n = 2802) among the in-school AGYW.

## Sexual debut experiences among AGYW

The overall mean age at first sex among AGYW interviewed was 16.6 years (Table 6), 16.2 years for in-school and 16.8 years for out of school.

Overall, most of the respondents had first time sex with either their boyfriend (80.4%) or husband (15.6%), with 2.4% of the in school reported to have had their sexual debut in marriage compared to 21.0% among out of school AGYW. Conversely, 91.6% of in-school reported to have had their sexual debut with boyfriends compared to 75.9% among the out of school AGYW. This means that overall, only 4% of respondents had their sexual debut with someone other than their boyfriend or husband. Table 6 also shows the age of the partner the AGYW respondents had their sexual debut with. Overall, partner age ranges 1–2 and 3–4 years older and AGYW interviewed, this age group constituted the majority of their partners at sexual debut (31.3% and 30.2%, respectively). The in school AGYW were more likely to have a partners of age range 1–2 years compared to their out of school counterparts (38.3% vs.28.5%, respectively) at sexual debut. Few respondents from both in-school and out of school (3.24% vs. 3.42%, respectively) had their partners who were 10 or more years at sexual debut.

**Willingness to have sex for the first time.** Table 6 also shows that 18.0% of AGYW were not willing at all to have sex the first time they were with their partner but 69.6% were very willing. There was a significant difference (p-value<0.0001) between the in-school and the out of school with regards to both questions; very willing (64.2% vs.71.9%, respectively) and not willing at all (22.1% vs.16.3%) respectively.

**Pregnancy prevention at first sex.** Table 6 shows that the primary method of pregnancy prevention at sexual debut among the study respondents was a male condom (92.7%). There was no significant difference between the two groups, the in-school at 93.5% while the out of school were at 92.3%. Overall female condom utilisation at first sexual debut was less than 1%. The practicality of it is of course questionable. Barrier methods have a dual protection aspect, HIV and pregnancy. Across all the different methods assessed among our participants, there were no significant differences between the in-school and the out of school respondents. Those reporting a withdrawal method as a method to prevent pregnancy accounted for 1.24% of the AGYW interviewed. This method clearly points to absence of barrier during sexual intercourse which is a risk factor for HIV acquisition

## Number of sexual partners

**Self-reported number of sexual partners.** From Table 7, the AGYW who had ever had sex were asked about the number of sexual partners that they had sex with in the last 12

**Table 6. Distribution of first-time sexual behaviour experiences among the participants, overall and by school status.**

| Characteristics | Total N = 12, 813 | In-School N = 5979 (46.7%) | Out-of-School N = 6834 (53.3%) | p-value <0.0001 |
|---|---|---|---|---|
| **Overall** | 12, 813 (100) | 5979 (100) | 6834 (1000 | n/a |
| **Ever had any sexual intercourse in life** | 8,406 (65.6%) | 2,436 (40.7%) | 5,970 (87.4%) | <0.0001 |
| Age at first sex, mean (SD)* | 16.6 (2.1) | 16.2 (2.1) | 16.8 (2.1) | <0.0001[T] |
| **Person with whom participant had sex for the first time*** | | | | |
| Boyfriend | 6,759 (80.4%) | 2,231 (91.6%) | 5,528 (75.9%) | <0.0001 |
| Husband | 1,313 (15.6%) | 59 (2.42%) | 1,254 (21.0%) | <0.0001 |
| Stranger | 53 (0.6%) | 24 (0.99%) | 29 (0.45%) | 0.813 |
| Just a friend or classmate | 203 (2.4%) | 85 (3.49%) | 118 (1.98%) | 0.506 |
| Brother | 2 (0.02%) | 2 (0.08%) | 0 (25.5%) | n/a |
| Teacher | 8 (0.1%) | 0 (0.0%) | 8 (0.13%) | n/a |
| Uncle | 10 (0.1%) | 6 (0.25%) | 4 (0.07%) | 0.915 |
| Father | 3 (0.04%) | 1 (0.04%) | 2 (0.03%) | n/a |
| Other relative | 24 (0.29%) | 11 (0.45%) | 13 (0.22%) | 0.922 |
| Other person | 31 (0.37%) | 17 (0.70%) | 14 (0.23%) | 0.852 |
| **Age of the partner the participant had sex with the first time*** | | | | |
| Same age | 503 (6.00%) | 183 (7.50%) | 320 (5.36%) | 0.336 |
| Younger | 40 (0.48%) | 12 (0.49%) | 28 (0.47%) | 0.993 |
| 1–2 years older | 2,632 (31.3%) | 932 (38.3%) | 1,700 (28.5%) | <0.0001 |
| 3–4 years older | 2,539 (30.2%) | 704 (28.9%) | 1,835 (30.7%) | 0.377 |
| 5–9 years older | 1,412 (16.8%) | 297 (12.2%) | 1,115 (18.7%) | 0.009 |
| 10 or more years older | 283 (3.37%) | 79 (3.24%) | 204 (3.42%) | 0.940 |
| Don't know | 997 (11.9%) | 229 (9.40%) | 768 (12.9%) | 0.154 |
| **Willingness to have sex at the first time** | | | | |
| Very willing | 5, 854 (69.6%) | 1,564 (64.2%) | 4,290 (71.9%) | <0.0001 |
| Somewhat willing | 950 (11.3%) | 303 (12.4%) | 647 (10.8%) | 0.468 |
| Not willing at all | 1,510 (18.0%) | 538 (22.1%) | 972 (16.3%) | 0.005 |
| Don't know | 92 (1.09%) | 31 (1.27%) | 61 (1.02%) | 0.914 |
| **Did something to prevent pregnancy at the first sexual intercourse. Method used to avoid pregnancy at the first sexual intercourse.** | N = 2,734 (21.3%) | N = 1,024 (17.1%) | 1,710 (25.0%) | |
| Male condom | 2,535 (92.7%) | 957 (93.5%) | 1,578 (92.3%) | 0.259 |
| Pill | 61 (2.23%) | 19 (1.86%) | 42 (2.46%) | 0.884 |
| Injection | 52 (1.90%) | 18 (1.76%) | 34 (1.99%) | 0.954 |
| Female condom | 24 (0.88%) | 10 (0.98%) | 14 (0.82%) | 0.967 |
| Withdrawal | 34 (1.24%) | 5 (0.49%) | 29 (1.70%) | 0.831 |
| Emergency contraception | 33 (1.21%) | 20 (1.95%) | 13 (0.76%) | 0.782 |
| IUD/Coil | 4 (0.15%) | 1 (0.10%) | 3 (0.18%) | n/a |
| Implant | 9 (0.33%) | 3 (0.29%) | 6 (0.35%) | n/a |
| Others | 13 (0.48%) | 4 (0.395) | 5 (0.39%) | n/a |
| **Was under the influence of drugs or alcohol at the time of first sex** | | | | |
| Yes | 368 (4.38%) | 94 (3.86%) | 274 (4.59%) | 0.766 |
| No | 7,885 (93.8%) | 2,291 (94.1%) | 5,594 (93.7%) | 0.503 |
| Don't know/Don't remember | 153 (1.82%) | 51 (2.09%) | 102 (1.71%) | 0.869 |

& = z-test comparing proportions of in-school vs out-of-school

* = analysis done on complete case data excluding the missing/non-response values, T = t test comparing in-school vs out-of-school, n/a = not applicable, due to small numbers.

**Table 7. Number of sexual partners by school status and age-group.**

|  |  | Number of sexual partners in the past 12 months | | |
|---|---|---|---|---|
|  |  | 0 | 1 | 2+ |
| School status |  |  |  |  |
| In-school | N = 1,892 | 146 (7.72%) | 1,415 (74.8%) | 331 (17.5%) |
| Out-of-school | N = 4,933 | 424 (10.6%) | 3,563 (72.2%) | 846 (17.2%) |
| Age groups |  |  |  |  |
| 10–14 years | N = 85 | 10 (11.8%) | 71 (83.5%) | 4 (4.71%) |
| 15–19 years | N = 1,860 | 140 (7.53%) | 1,426 (76.7%) | 294 (15.8%) |
| 20–24 years | N = 4,880 | 520 (10.7%) | 3,481 (71.3%) | 879 (18.0%) |
| Total | N = 6,825 | 670 (9.82%) | 4,978 (72.9%) | 1,177 (17.3%) |

months and in their lifetime. For sex within the last 12 months, overall, the majority of the AGYW 72.9% (n = 4978) reported engaging in sex with one (1) sexual partner in the past 12 months, with minimal differences between the in-school 74.8% (n = 1415) and the out-of-school 72.2% (n = 3563). However, 17.3% (n = 1177) AGYW reported engaging in sex with 2 + partners in the past 12 months. By age-group, the proportion of AGYW reporting having one sexual partner in the past 12 months was decreasing slightly with increasing age. That is, 83.5% (n = 71) for the 10–14 years, 76.7% (n = 1426) for the 15 to 19 years, and 71.3% (n = 3481) for the 20 to 24 years. On the other hand, the proportion reporting 2+ partners in the past year increased with increasing age with 4.7% (n = 4) for the 10 to '4 years, 15.8% (n = 294) for the 15 to 19 years and 18% (n = 879) for the 20 to 24 years. Only 9.8% (n = 670) reported having no sexual partner in the past 12 months.

## Multiple sexual partnerships, condom use at last sex and consistent condom use by district of residence

**Multiple sexual partnerships.** Table 8 shows the percentage of respondents that reported multiple sexual partners, condom use at last sex and consistent condom use by district of residence. The proportion of those reporting 2 or more sexual partners was highest in Kalomo (60.9%), Nyimba (39.6%) and Mwense (36.4%). Districts with the lowest proportion of AGYW reporting 2+ sexual partners in the past 12 months include Chililabombwe (6.5%), Mungwi (9.3%) and Mkushi (9.9%). There is a need to focus on high-risk districts (with the highest proportions reporting 2+ sexual partners) with targeted HIV prevention efforts to reduce AGYW's vulnerability to HIV infection.

**Condom use at last sex and consistent condom use.** Condom use at last sex was generally low across districts (Table 8). The proportion of AGYW reporting condom use at last sex was highest in Mongu (46.1%), Kabwe (43.9%) and Kalomo (41.2%). The proportion was low in Chembe (12.9%), Petauke (18.7%) and Sinazongwe (18.4%). The highest proportion of consistent condom users reported in Kalomo (24.2%), Mongu (22.9%), and Limulunga (21.8%) in that order. The following are the lowest consistent condom use: Chembe (4.1%), Chinsali (4.4%), Petauke (6.4%) and Sinazongwe (7.5%). The proportions reporting condom use at last sex or consistent condom use are still too low to impact on HIV transmission suggesting a need to reinforce condom use messages among AGYW across all districts.

## Prevalence and factors associated with hiv and syphilis

**Overall prevalence of HIV and syphilis.** Out of 12, 457 participants the overall prevalence of HIV was 466 (3.7%) 95% CI: 3.4–4.1). The weighted prevalence of HIV was higher in

**Table 8. Percentage of respondents that had sex in the past 12 months, and of these, percentage that reported multiple sexual partners, condom use at last sex and consistent condom use by district of residence.**

| District | Percentage of respondents who reported that they had sex with any sexual partners in the past 12 months | Of those that had sex in the past 12 months, percentage that reported that they: | | | |
|---|---|---|---|---|---|
| | | N | Had sex with 2 or more sexual partners | Used a condom at last sex with most recent sexual partner | Used a condom all the time they had sex with their sexual partners (consistent condom use) |
| Chembe | 88.0 | 411 | 21.4 | 12.9 | 4.1 |
| Chililabombwe | 78.6 | 261 | 6.5 | 25.7 | 8.4 |
| Chinsali | 87.6 | 429 | 11.9 | 29.1 | 4.4 |
| Chisamba | 88.1 | 384 | 20.8 | 26.3 | 12.7 |
| Chongwe | 90.3 | 307 | 16.6 | 33.9 | 19.2 |
| Kabwe | 79.5 | 330 | 21.8 | 43.9 | 18.5 |
| Kalomo | 87.5 | 405 | 60.9 | 41.2 | 24.2 |
| Kalulushi | 77.7 | 342 | 17.3 | 29.8 | 16.1 |
| Kasama | 81.4 | 319 | 19.7 | 22.6 | 10.0 |
| Limulunga | 72.4 | 239 | 13.4 | 42.7 | 21.8 |
| Mazabuka | 91.9 | 440 | 16.4 | 40.2 | 16.4 |
| Mkushi | 84.1 | 391 | 9.9 | 24.3 | 10.7 |
| Mungu | 84.5 | 399 | 21.8 | 46.1 | 22.9 |
| Mungwi | 83.7 | 247 | 9.3 | 23.1 | 10.9 |
| Mwense | 91.7 | 440 | 36.4 | 23.4 | 12.7 |
| Mwinilunga | 86.4 | 424 | 15.3 | 33.7 | 10.6 |
| Ndola | 84.4 | 306 | 10.8 | 30.7 | 18.6 |
| Nyimba | 93.8 | 452 | 39.6 | 19.0 | 7.5 |
| Petauke | 90.9 | 454 | 25.1 | 18.7 | 6.4 |
| Sinazongwe | 91.8 | 267 | 25.2 | 18.4 | 7.5 |

the out-of-school compared to the compared to the in-school participants (5.5% vs 2.0%), Similarly, the prevalence of syphilis was higher in the out-of-school than the in-school participants (4.1% vs 1.5%). When the HIV prevalence was stratified by age group, highest prevalence was reported among those aged 20–24 years (5.8%) while the least was among those aged 10–14 years (0.7%). Likewise, highest prevalence of syphilis was reported among participants aged 20–24 years (4.7%) and lowest was among those aged 10–14 years (0.5%) as shown in Table 9.

**Prevalence of HIV and syphilis by district.** Fig 2 shows that Mkushi and Mungu districts had the highest HIV prevalence of 641 (6.7%) and 640 (6.7%)% respectively, followed by Mwense 691 (5.6%), Kabwe 655 (5.2%) in that order. Districts that reported least prevalence of

**Table 9. HIV and syphilis prevalence among AGYW, by school status and age-group.**

| N = 12813 | | Unweighted | | Weighted | |
|---|---|---|---|---|---|
| | | HIV % (95% CI) | Syphilis % (95% CI) | HIV % (95% CI) | Syphilis % (95% CI) |
| **School status** | | | | | |
| In-school | **5,979** | 2.1%(1.8%, 2.5%) | 1.7% (1.4%, 2.0%) | 2.0%(1.9%, 2.3%) | 1.5%(1.4%, 1.7%) |
| Out-of-school | **6,834** | 5.3%(4.8%, 5.9%) | 4.6% (4.1%, 5.1%) | 5.5%(5.3%, 5.7%) | 4.1%(3.9%, 4.2%) |
| **Age groups** | | | | | |
| 10–14 years | **1,617** | 0.7%(0.3%, 1.2%) | 0.5% (0.2%, 1.0%) | 0.7%(0.6%, 0.8%) | 0.5% (0.4, 0.6%) |
| 15–19 years | **4,536** | 2.4%(1.9%, 2.9%) | 1.6% (1.2%, 2.0%) | 2.7%(2.5%, 2.9%) | 1.4%(1.3%, 1.5%) |
| 20–24 year | **6,660** | 5.5%(5.0%, 6.1%) | 5.1% (4.5%, 5.6%) | 5.8%(5.6%, 6.0%) | 4.7% (4.7, 4.9%) |

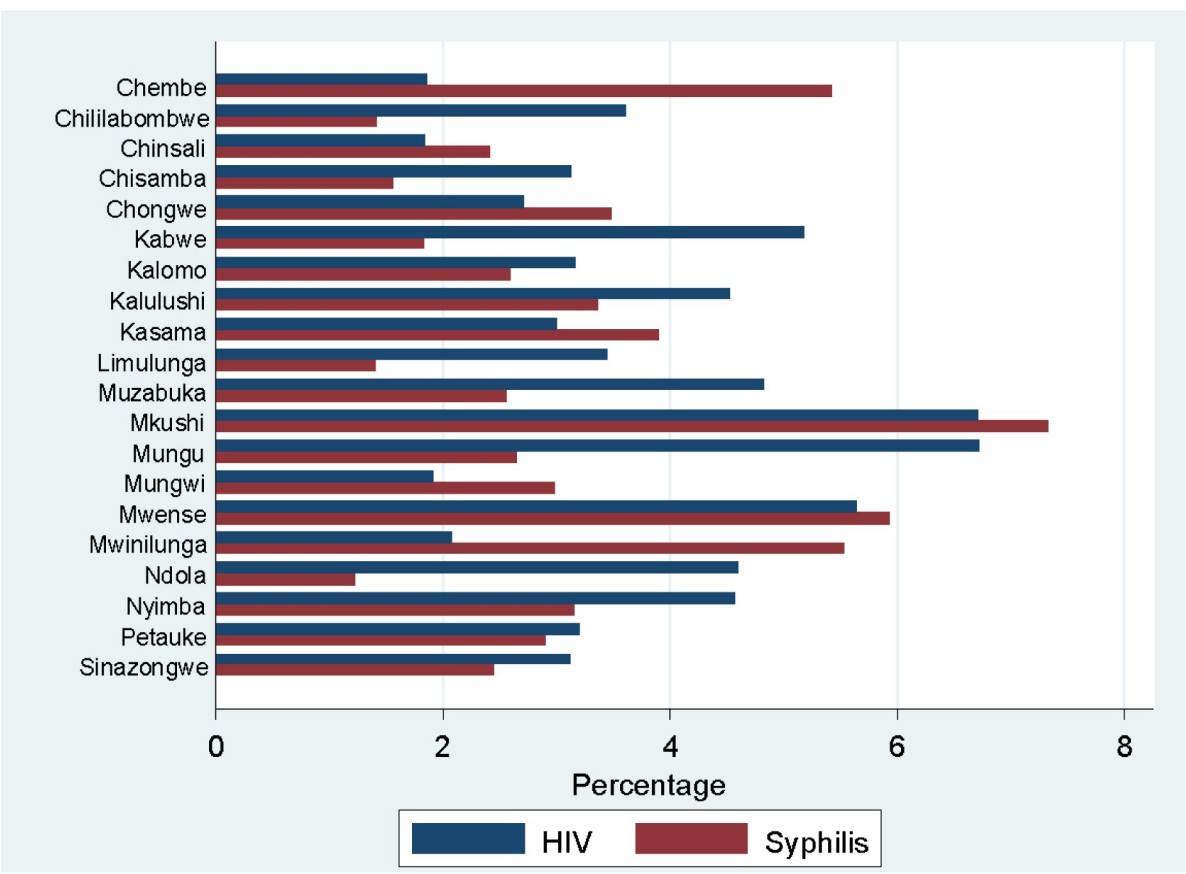

**Fig 2. Prevalence of HIV and syphilis by districts.**

HIV were Chinsali 705 (1.8%), Chembe 701 (1.9%), Mungwi (1.9%) and Mwinilunga 723 (2.1%). For syphilis, the districts that reported highest prevalence were Mkushi 641 (7.3%), Mwense 691 (5.9%), Mwinilunga 723 (5.5%) and Chembe 701 (5.4%). Least prevalence of syphilis was reported from Ndola 652 (1.2%), Chililabombwe 636 (1.4%), Limulunga 639 (1.4%) and Chisamba 703 (1.65%). The overall prevalence of syphilis was 12, 813 (3.2%).

**Factors associated with HIV infection.** In the bivariable logistic regression analysis for factors associated with HIV infection among AGYW in selected districts of Zambia: age, ever married, school status, ever had sex, knowledge of HIV prevention and vulnerability were associated with HIV infection. However, in the multivariable regression, only age, ever had sex and school status were significantly associated with HIV infection. AGYW who were aged 15–19 years were 2.25 (1.17–4.31) times the odds of HIV infection compared to their counterparts who were aged 10–14 years. Similarly, those who were aged 20–24 years were 3.67 (1.89–7.11) times the odds of HIV infection compared to those who were aged 10–14 years. Being out-school was associated with 1.40 (1.10–1.78) times the odds of HIV infection compared to those who were in-school. Also, this study examined that the likelihood of having HIV infection was 1.84 fold (1.34–2.51) higher among AGYW who reported 'ever to have sex' compared to those who had 'never experienced sex' (Table 10).

**Factors associated with syphilis infection.** Regarding syphilis infection, the bivariable logistic regression analysis showed age, ever married, school status, ever had sex, knowledge of HIV prevention and vulnerability were associated with syphilis infection.

**Table 10. Bivariable and multivariable logistic regression analysis for factors associated with HIV infection among AGYW in selected districts of Zambia, 2021.**

| Characteristic | Bivariable | | Multivariable | |
|---|---|---|---|---|
| | COR (95% CI) | P- value | AOR (95% CI) | P-value |
| Age (years) | | | | |
| 10–14 | Ref | | | |
| 15–19 | 3.53 (1.92–6.69) | <0.001 | 2.25 (1.17–4.31) | 0.014 |
| 20–24 | 8.51 (4.66–15.5 | <0.001 | 3.67 (1.89–7.11) | <0.001 |
| Marital status | | | | |
| Not Married | Ref | | | |
| Married | 1.91 (1.58–2.29) | <0.001 | 1.05 (0.84–1.29) | 0.654 |
| School status | | | | |
| In-school | Ref | | | |
| Out-of-school | 2.56 (2.08–3.14) | <0.001 | 1.40 (1.10–1.78) | 0.005 |
| Ever had sex | | | | |
| No | Ref | | | |
| Yes | 3.66 (2.81–4.79) | <0.001 | 1.84 (1.34–2.51) | <0.001 |
| Knowledge of HIV | | | | |
| Low | Ref | | | |
| High | 0.53 (0.42–0.66) | <0.001 | 0.79 (0.53–1.15) | 0.229 |
| Vulnerability | | | | |
| Low | Ref | | | |
| Medium | 1.86 (1.42–2.41) | <0.001 | 1.03 (0.68–5.92) | 0.88 |
| High | 1.48 (1.03–2.21) | 0.034 | 0.86 (0.59–1.45) | 0.59 |

COR = Crude odds ratio; AOR = adjusted odds ratio; CI = confidence interval Ref = reference category; HIV = Human immunodeficiency virus

However, in the multivariable regression only, age, ever married and ever had sex were significantly associated with syphilis infection. Participants who were aged 20–24 years were 3.11 (95% CI,1.41–6.73) times greater odds of having syphilis infection compared to those aged 10–14 years. For marriage, those who reported that they were ever married were 1.26 (95% CI, 1.01–1.63) times the odds of having syphilis infection compared to those who reported that they were never married. Similarly, participants who indicated that they had ever experienced sexual intercourse were 3.41(95% CI, 2.23–5.38) times the odds of syphilis infection compared to their counterparts who were reported not to have ever experienced sexual intercourse (Table 11).

## Discussion

In what follows, we discuss our findings with respect to: *1) Behavioural characteristics of AGYW, 2) Prevalence of HIV and Syphilis and associated factors, and 3) Vulnerability levels of households and individual AGYYW.*

### Behavioural characteristics of AGYW

Our findings in this study suggest that several sexual and health behavioural challenges and needs of AGYW need to be addressed by different players at different levels of society. It's important to note that in some of the behavioural aspects reported in this study the out-of-school AGYW are much more disadvantaged than their in-school AGYW. For example, the proportion of AGYW that had never had sex were 87.4% among the in school and 40.7% among the out of school AGYW. Also concerning willingness to have sex at the first time,

**Table 11. Bivariable and multivariable logistic regression analysis for factors associated with syphilis infection among AGYW in selected districts of Zambia, 2021.**

| Characteristic | Bivariable | | Multivariable | |
|---|---|---|---|---|
| | COR (95% CI) | P- value | AOR (95% CI) | P-value |
| Age (years) | | | | |
| 10–14 | Ref | | | |
| 15–19 | 3.19 (1.53–6.65) | 0.002 | 1.44 (0.66–3.13) | 0.356 |
| 20–24 | 10.6 (5.27–21.5) | <0.001 | 3.11 (1.41–6.73) | 0.005 |
| Marital status | | | | |
| Not Married | Ref | | | |
| Married | 2.73 (2.21–3.38) | <0.001 | 1.26 (1.01–1.63) | 0.047 |
| School status | | | | |
| In-school | Ref | | | |
| Out-of-school | 2.79 (2.22–3.51) | <0.001 | 1.14 (0.87–1.48) | 0.317 |
| Ever had sex | | | | |
| No | Ref | | | |
| Yes | 6.74 94.66–9.74) | <0.001 | 3.41 (2.23–5.38) | <0.001 |
| Knowledge of HIV | | | | |
| Low | Ref | | | |
| High | 1.59 (1.26–2.01) | <0.001 | 0.73 (0.51–1.05) | 0.090 |
| Vulnerability | | | | |
| Low | Ref | | | |
| Medium | 1.78 (1.3 4–2.35) | <0.001 | 1.48 (0.97–2.28) | 0.068 |
| High | 1.58 (1.07–2.310 | 0.019 | 1.48 (0.87–2.52) | 0.139 |

COR = Crude odds ratio; AOR = adjusted odds ratio; CI = confidence interval Ref = reference category

71.9% out of school compared to 64.2% in school AGYW were very willing to have sex. This finding is in the same direction as found by Matovu et al., (2021) [11]. In addition, 40.2% out-of-school AGWY reported having sex within a week before the survey compared to only 13.4% of the in-school. Also, concerning use of condoms, 45.9% in-school AGYW used a condom the last time they had sex compared to 23.6% of the out-of-school. However, consistent use of condoms in the past 12 months was very low in both groups; 10.1% and 11.2% in-school and out-of-school AGYW respectively. Concerted effort is needed on the out-of-school AGYW since they seem to be much more disadvantaged than the in-school AGYW. Stakeholders at different levels who are currently working with AGYW need to continue with and increase the STI behavioural education that promote behavioural change in this particular group, especially among the out-of-school AGYW.

## Prevalence of HIV and syphilis and associated factors

The overall HIV prevalence was 3.7%, with a higher prevalence among out-of-school adolescents (as compared to the in-school adolescents (5.5%) compared to the in-school adolescents (2.0%). This finding is consistent with other studies [8, 12, 28–30]. Similarly, the prevalence of syphilis was higher in the out-of school than the in-school participants (4.1% vs 1.5%). The study also revealed some gaps in knowledge that existed among the AGYW. For example, 23% of AGYW did not know that a healthy-looking person can have HIV; 40% did not know that HIV can be transmitted from mother to child during pregnancy; and 47% thought if a man/woman had HIV the partner will also have HIV. Therefore, there is a need to scale up efforts that target AGYW especially the out of school AGYW with interventions to address HIV and

Syphilis amongst this group. Health promotion and education interventions can be used to enhance targeted messages to address knowledge gaps about HIV. In addition, proven high impact biomedical interventions such as PreP remain key to addressing the HIV risk at individual level for the AGYWs.

## Vulnerability levels of households and individual AGYW

Findings show that at individual level the majority of out-of-school girls (71.6%) were categorized as highly vulnerable compared to the in-school girls (69.7%). Having ever given birth, first sex before the age of 15 years and having multiple sexual partners were indicative of very high vulnerability among the girls. At household level, all indicators of vulnerability had high proportions of girls in the high tertile. Community vulnerability showed a similar trend. Drivers of vulnerability include living with parents, pregnant or given birth, having sex before 15 years and with multiple sexual partners, at individual level. At household level, the head of the household without education and lack of access to improved sanitation or water were indicative of high vulnerability. Similarly, at community level AGYW who were HIV positive, and knowledge about HIV were indicative of high vulnerability. The high levels of individual, household and community-level vulnerabilities suggest that interventions are needed to reduce the levels of vulnerabilities.

The vulnerabilities of AGYW observed also contributes to the observed high prevalence in AGYW compared to their men counterpart. In addition, girls' intimate relationships and sexual debut is earlier than their male peers; and secondly, they have partners older than them, some of whom emanate from age groups/brackets with higher HIV incidence. The power dynamics in these relationships usually leave the female folk vulnerable, especially amidst questionable condom use and/or its absence and lack of other personalized, and female tailored HIV prevention interventions such as the Cabotegravir injectable and Dapivirine ring, respectively. Both of which are only being introduced now. There is also a biological part which makes it easier for HIV acquisition in these young females. Of course, in our setting, there is the issue of Sexual Gender Based Violence (SGBV) which is a big problem in Zambia. Our study showed that all these factors contributed to the fact that HIV prevalence was high among AGYW compared to men.

## Sexual reproductive services in Zambia

The SRH services available in Zambia are: Family planning (contraceptive) services, antenatal, postnatal, delivery, safe and post abortion care, cervical cancer screening, HIV testing and treatment services, Human Papilloma Virus Vaccination for pre and adolescent girls aged between 9 and 14, menstrual hygiene and management. These services are available and accessible at different levels of the health care system from primary to tertiary level and ranging from information only, counselling, and actual services.

However, the SRH services availability can be improved by ensuring availability of SRH commodities and equipment, trained personnel, and right infrastructure for provision of comprehensive SRH services.

## Conclusion

In conclusion, the findings suggest that there are differences in sexual-risk behaviours and the prevalence of HIV and syphilis between in-and out-school AGYW. These differences are such that, out-of-school AGYW tended to have high prevalence of both HIV and Syphilis compared to in-school AGYW. Further, in-school AGYW were significantly less likely to engage in sex at an early age, and when they eventually did, they were more likely to engage in protected sex

than their out-of-school counterparts. With respect to the districts, the findings show that particular attention needs to be taken in Mkushi, Mwense and Chembe (for Syphilis). These districts stood out in having high prevalences of HIV and Syphilis.

## Recommendations

These findings support the view that there is a need for interventions that can help to keep girls in school, provide training for teachers on how to deliver comprehensive sexual education (CSE) effectively, engage with community leaders and parents to build support for CSE implementation and Increase funding and resources for implementing CSE programmes including providing materials and equipment for teaching. Among those AGYW already out of school, there is a need for unique interventions to reduce risk taking behaviours such as improving on their ability to negotiate for safer sex, and reduce their vulnerability to the risk of HIV and other STIs.

## Acknowledgments

We would like to express our heartfelt gratitude to all the individuals and institutions who supported and contributed towards the implementation of this project. Without their valuable support, this project would not have been feasible. We are especially grateful to the Ministry of Health and Global Fund through the Prevention of HIV, TB, and Malaria for providing financial and technical support for this project. We are also further grateful to the Ministry of Health Leadership through the Adolescent Health Unit for their oversight and guidance throughout the study. Their expertise and support were vital in ensuring that the research adhered to the highest standards of ethics and safety.

We would also like to extend our thanks to CIDRZ for their significant contribution to the success of this project. Their facilitation of testing of samples and maintaining quality controls were critical in ensuring the accuracy and reliability of the data.

We also extend our sincere thanks to the Ministry of Health, Education Government Departments, Non-Governmental Organisations, Community Radio Stations, and Zambia National Broadcasting Corporation for their support during the data collection process. We also extend our appreciation to the District Health Promotion and Adolescent Health Officers for their tireless efforts towards sensitization and data collection. Their dedication and commitment to promoting HIV testing and counselling for adolescent girls and young women in Zambia is commendable.

Additionally, we extend our gratitude to the research assistants, laboratory scientists, and counsellors for their valuable contributions to this project. Their hard work and dedication towards data collection and analysis were crucial to the success of this research.

Furthermore, we would also like to extend our gratitude to all the investigators who played a crucial role in this study. Their expertise, guidance, and support were vital to the success of this research project. Finally, we would also like to express our gratitude to the project management team for their tireless efforts in overseeing the day-to-day activities of this project. Their efficient planning and execution of the project ensured that the research was completed within the stipulated timeframe and budget.

Finally, we would like to acknowledge the traditional, religious leaders and parents in the various study sites for their support and guidance. Their involvement was instrumental in ensuring that the research was culturally sensitive and relevant to the local community. Special thanks to all those who agreed to participate in the study, as without them, this report could not have been possible.

## Author Contributions

**Conceptualization:** Patrick Musonda, Hikabasa Halwiindi, Patrick Kaonga, Alice Ngoma-Hazemba, Chowa Tembo, Cosmas Zyambo, Owen Ngalamika, Mwiche Musukuma, Malizgani P. Chavula, Noah Sichula, Oliver Mweemba, Joseph Mumba Zulu, Henry Phiri.

**Data curation:** Patrick Musonda, Hikabasa Halwiindi, Patrick Kaonga, Alice Ngoma-Hazemba, Mable Mweemba, Chowa Tembo, John Chisoso, Margaret Munakampe, Powell Choonga, Owen Ngalamika, Mwiche Musukuma, Malizgani P. Chavula, Oliver Mweemba, Joseph Mumba Zulu.

**Formal analysis:** Patrick Musonda, Patrick Kaonga, Alice Ngoma-Hazemba, Matildah Simpungwe, Mable Mweemba, Chowa Tembo, Margaret Munakampe, Owen Ngalamika, Mwiche Musukuma, Malizgani P. Chavula, Oliver Mweemba, Joseph Mumba Zulu.

**Funding acquisition:** Hikabasa Halwiindi, Cosmas Zyambo, Powell Choonga, Joseph Mumba Zulu, Henry Phiri.

**Investigation:** Patrick Musonda, Hikabasa Halwiindi, Matildah Simpungwe, Mable Mweemba, Chowa Tembo, Cosmas Zyambo, John Chisoso, Margaret Munakampe, Powell Choonga, Owen Ngalamika, Mwiche Musukuma, Malizgani P. Chavula, Noah Sichula, Oliver Mweemba, Joseph Mumba Zulu, Henry Phiri.

**Methodology:** Patrick Musonda, Hikabasa Halwiindi, Alice Ngoma-Hazemba, Mable Mweemba, Owen Ngalamika, Mwiche Musukuma, Noah Sichula, Oliver Mweemba, Joseph Mumba Zulu, Henry Phiri.

**Project administration:** Hikabasa Halwiindi, Cosmas Zyambo, John Chisoso, Margaret Munakampe, Powell Choonga, Mwiche Musukuma, Malizgani P. Chavula, Joseph Mumba Zulu, Henry Phiri.

**Resources:** Hikabasa Halwiindi, Matildah Simpungwe, Mwiche Musukuma, Joseph Mumba Zulu, Henry Phiri.

**Software:** Patrick Musonda, Patrick Kaonga, John Chisoso.

**Supervision:** Patrick Musonda, Margaret Munakampe, Powell Choonga, Mwiche Musukuma, Malizgani P. Chavula, Joseph Mumba Zulu, Henry Phiri.

**Validation:** Patrick Musonda, Hikabasa Halwiindi, Patrick Kaonga, Mable Mweemba, Chowa Tembo, John Chisoso, Owen Ngalamika, Mwiche Musukuma, Malizgani P. Chavula, Noah Sichula, Oliver Mweemba, Joseph Mumba Zulu, Henry Phiri.

**Visualization:** Patrick Musonda, Hikabasa Halwiindi, Cosmas Zyambo, Malizgani P. Chavula, Henry Phiri.

**Writing – original draft:** Patrick Musonda.

**Writing – review & editing:** Patrick Musonda, Hikabasa Halwiindi, Patrick Kaonga, Alice Ngoma-Hazemba, Mable Mweemba, Cosmas Zyambo, Joseph Mumba Zulu, Henry Phiri.

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
