## [Decision Letter · Decision Letter 0]

18 Jan 2024

PONE-D-23-35955HIV, Syphilis and Sexual-risk behaviours’ prevalence among in-and out-of-school adolescent girls and young women in Zambia: A cross-sectional survey studyPLOS ONE

Dear Dr. Musonda,

Thank you for submitting your manuscript to PLOS ONE. After careful consideration, we feel that it has merit but does not fully meet PLOS ONE’s publication criteria as it currently stands. Therefore, we invite you to submit a revised version of the manuscript that addresses the points raised during the review process. Please submit a revised version in line with the reviewer's queries and resubmit.

We look forward to receiving your revised manuscript.

Kind regards,

Moses Katbi

Academic Editor

PLOS ONE

“We would like to express our heartfelt gratitude to all the individuals and institutions who supported and contributed towards the implementation of this project. Without their valuable support, this project would not have been feasible. We are especially grateful to the Ministry of Health and Global Fund through the Prevention of HIV, TB, and Malaria for providing financial and technical support for this project. We are also further grateful to the Ministry of Health Leadership through the Adolescent Health Unit for their oversight and guidance throughout the study. Their expertise and support were vital in ensuring that the research adhered to the highest standards of ethics and safety.

We would also like to extend our thanks to CIDRZ for their significant contribution to the success of this project. Their facilitation of testing of samples and maintaining quality controls were critical in ensuring the accuracy and reliability of the data.

We also extend our sincere thanks to the Ministry of Health, Education Government Departments, Non-Governmental Organisations, Community Radio Stations, and Zambia National Broadcasting Corporation for their support during the data collection process. We also extend our appreciation to the District Health Promotion and Adolescent Health Officers for their tireless efforts towards sensitization and data collection. Their dedication and commitment to promoting HIV testing and counselling for adolescent girls and young women in Zambia is commendable.

Additionally, we extend our gratitude to the research assistants, laboratory scientists, and counsellors for their valuable contributions to this project. Their hard work and dedication towards data collection and analysis were crucial to the success of this research.

Furthermore, we would also like to extend our gratitude to all the investigators who played a crucial role in this study. Their expertise, guidance, and support were vital to the success of this research project. Finally, we would also like to express our gratitude to the project management team for their tireless efforts in overseeing the day-to-day activities of this project. Their efficient planning and execution of the project ensured that the research was completed within the stipulated timeframe and budget.

Finally, we would like to acknowledge the traditional, religious leaders and parents in the various study sites for their support and guidance. Their involvement was instrumental in ensuring that the research was culturally sensitive and relevant to the local community. Special thanks to all those who agreed to participate in the study, as without them, this report could not have been possible.

“This study was funded by the Global Fund. The Zambia AGYW program is part of the Global Fund Strategy (2017-2022) to reduce new HIV infections among AGYW by 58% by 2022 in13 sub-Saharan African countries including Botswana, Cameroon, Kenya, Lesotho, Malawi, Mozambique, Namibia, South Africa, Eswatini, Tanzania, Uganda, Zambia and Zimbabwe. However, the funders played no part in writing of this manuscript, any errors or mistakes are thoroughly the responsibility of the authors.”

3. In the online submission form you indicate that your data is not available for proprietary reasons and have provided a contact point for accessing this data. Please note that your current contact point is a co-author on this manuscript. According to our Data Policy, the contact point must not be an author on the manuscript and must be an institutional contact, ideally not an individual. Please revise your data statement to a non-author institutional point of contact, such as a data access or ethics committee, and send this to us via return email. Please also include contact information for the third party organization, and please include the full citation of where the data can be found.

Reviewers' comments:

Reviewer's Responses to Questions

**Comments to the Author**

1. Is the manuscript technically sound, and do the data support the conclusions?

Reviewer #1: Yes

2. Has the statistical analysis been performed appropriately and rigorously? 

Reviewer #1: Yes

3. Have the authors made all data underlying the findings in their manuscript fully available?

Reviewer #1: No

4. Is the manuscript presented in an intelligible fashion and written in standard English?

Reviewer #1: Yes

5. Review Comments to the Author

Reviewer #1: HIV, Syphilis and Sexual-risk behaviours’ prevalence among in-and out-of-school adolescent girls and young women in Zambia: A cross-sectional survey study is a well written paper that has provided in depth information in the introduction, material and methods and results sections. This is to be commended. The paper has brought out very well the differences between the in school and out of school AGYW groups and the high level of vulnerability that both groups have in terms of acquiring HIV and syphilis . The Discussion section, however is very brief and inadequate compared to the wealth of information that has been provided in all the other sections . I would suggest reducing the content in the materials and methods section (especially the formulas and the details in assessing the sample size- a few sentences on how the sample size was arrived at would suffice).It would be good if the discussion could include the following :

1. Why do AGYW in Zambia and other developing nations have a much higher incidence of HIV when compared to the men(5.6%vs 1.8%)

2. The authors have mentioned the seroprevalence of HIV and Syphilis in the groups district wise. It would be nice to know how many were positive for both HIV and Syphilis , if this data is available.

3. One of the objectives of the study as mentioned in Introduction was to determine the incidence of GBV in these groups of AGYW. However nothing is mentioned about GBV in the discussion.

4. What are the SRH services (or lack of it) available in Zambia and how this can be improved as SRH services availability was also one of the objectives of the study.

5. What suggestions do the authors have to improve the situation of AGYW in Zambia and decrease their vulnerability.

6. would suggest that the authors number the lines in their manuscript.

6. PLOS authors have the option to publish the peer review history of their article (what does this mean?). If published, this will include your full peer review and any attached files.

Reviewer #1: No

---

## [Author Response · Author response to Decision Letter 0]

19 Mar 2024

We have responded to comments in the attached response to reviewers.

---

## [Editor Report · Decision Letter 1]

7 May 2024

HIV, Syphilis and Sexual-risk behaviours’ prevalence among in-and out-of-school adolescent girls and young women in Zambia: A cross-sectional survey study

PONE-D-23-35955R1

Dear Prof Musonda Patrick,

We’re pleased to inform you that your manuscript has been judged scientifically suitable for publication and will be formally accepted for publication once it meets all outstanding technical requirements.

Kind regards,

Moses Katbi

Academic Editor

PLOS ONE
---

## [Editor Report · Acceptance letter]

24 May 2024

PONE-D-23-35955R1 

PLOS ONE

Dear Dr. Musonda, 

I'm pleased to inform you that your manuscript has been deemed suitable for publication in PLOS ONE. Congratulations! Your manuscript is now being handed over to our production team.

Kind regards, 

on behalf of

Dr. Moses Katbi 

Academic Editor

PLOS ONE